# Not All Imbalance Is Random: Cluster-Balanced Ensembling for Missing-Not-At-Random Class Imbalance

## Abstract

Class imbalance methods inherently assume that observed minority instances are representative of their class and Missing At Random (MAR). However, in many real-world settings, minority instances are Missing Not At Random (MNAR), with observability shaped by both class and feature values. This leads to structurally biased samples, introducing a deeper challenge that goes beyond class-count imbalance. We show that when MNAR affects high-impact features, popular imbalance methods overfit the observed minority and fail to generalize. To address this, we propose a simple yet effective cluster-balanced ensemble approach that constructs diverse, near-balanced training sets by pairing all minority instances with different clusters of the majority class. Extensive experiments identify MNAR conditions under which our approach improves F1 scores over existing methods, and when it does not. We also introduce an evaluation protocol using representative balanced test sets, demonstrating that standard hold-out testing on MNAR data can mislead performance assessments. Our findings underscore that the cause of imbalance is as critical as the correction method.

## 1 Introduction

Class imbalance is a prevalent real-world problem and well-known to distort supervised learning algorithms. When the distribution of classes is highly uneven, classification algorithms learn to prioritize the majority class and become poor detectors of the minority class. As such, they are susceptible to missing important patterns underlying rare instances. This is especially problematic in applications where the minority class is the primary focus of inference, for example, in customer churn, purchase conversion, fraud detection, and medical diagnosis.

Existing imbalance methods inherently assume observed minority instances provide an unbiased representation of their class population, and are Missing At Random (MAR). Under MAR, imbalance is mostly due to uneven number of instances, which can be effectively mitigated by re-balancing the dataset via various techniques. But in practice, observability often depends on both class labels and features, a condition known as Missing Not At Random (MNAR).

Under MNAR imbalance, certain regions of the minority class are disproportionately unobserved—beyond what their natural frequency in the data-generating distribution would suggest. This missingness occurs precisely due to their class and features values. For instance, the bankrupt firms we observe (minority) may not represent all bankruptcies, *since* small failed firms often vanish from records. This is unlike small non-bankrupt firms which remain visible. Similarly, observed defaulters (minority) may skew toward borrowers with fuller credit histories, excluding riskier profiles that are filtered out pre-loan.

The rare nature of the minority class makes MNAR especially probable in imbalanced datasets. Smaller samples of the population have inherently higher risk of underrepresenting or even excluding less common subpopulations, resulting in a biased representation of the minority population. Figure 1 exemplifies how the observed distribution of minority instances can differ under MAR and MNAR scenarios, imposing an additional challenge on the already difficult class imbalance problem. Our work exposes this overlooked bias and reframes imbalance correction through the lens of MNAR.

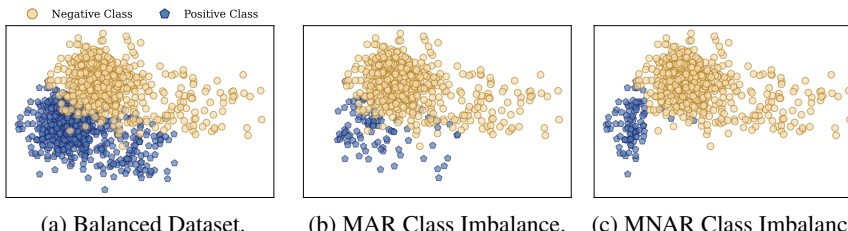

(a) Balanced Dataset.  (b) MAR Class Imbalance.  (c) MNAR Class Imbalance.

Figure 1: A balanced dataset with corresponding MAR and MNAR imbalance scenarios. Both imbalanced datasets have the same number of positive instances, but higher x-axis value instances are more likely to be missing under MNAR, making observed positive instances less representative of their class.

In the first step, we show that under higher MNAR imbalance (beyond 5:1 imbalance ratio), existing techniques can reinforce the very biases they aim to correct. In particular, they tend to overfit the observed instances in the minority class and generalizes poorly to the true population.

To address this, we next propose to exploit information from the majority class to uncover useful structures for handling MNAR imbalance in the minority class. In particular, if MNAR arises because certain feature regions are more likely to miss minority instances, then contrasting the minority group to diverse regions of the majority class (rather than to random samples) may provide unexplored benefits.

We show this by introducing a simple cluster-balanced ensemble approach, where a base classifier is trained on diverse, near-balance subsets of the input data. These subsets are constructed by clustering the majority class to specifically create segments that are of similar size to the entire minority class. Then, by pairing different segments with all minority instances, we expose the balanced ensemble to diverse decision boundaries, and show that it can better generalize to the true underlying minority distribution.

We compare our approach to state-of-the-art class imbalance techniques over several benchmark datasets and base classifiers, and consistently achieve higher F1 scores. We identify conditions under which the performance gap is most pronounced, and when it is not. For instance, we find that when MNAR is generally driven by features that highly affect AUC performance (particularly ROC AUC, but also PR AUC), have higher kurtosis, higher variance in the minority group, or most influence the underlying probability distribution of the minority class, our approach performs significantly better than existing methods. On the other hand, under MAR scenarios, or when MNAR is due to less predictive features, existing methods remain the best choice.

Lastly, we introduce a validation protocol for imbalance methods. Our protocol uses datasets that are originally near-balanced. Then, it simulates MAR imbalance by randomly deleting positive instances, or MNAR imbalance by deleting positive instances based on specific feature values. Imbalance methods are trained on the resulting damaged dataset, but validation is performed using hold-out sets from the original representative and balanced data.

The benefit of this protocol is that it allows us to evaluate imbalance methods by how they generalize to unobserved minority subgroups, which are potentially missing from MNAR imbalance data. As we show, this gives an unbiased evaluation compared to the standard approach of sampling the test set from the MNAR imbalanced data itself, which can lead to misleading assessments.

We note that our evaluation protocol relies on access to near-balanced datasets not typically available in real-world class imbalance scenarios. While this limits direct evaluation on naturally imbalanced datasets, our goal is not to propose a universally adoptable standard. Rather, we seek to highlight the potential pitfalls of conventional evaluation practices when class imbalance arises from MNAR.

Our contributions can be summarized into four parts:

- We present a conceptual and empirical examination of existing imbalance methods, demonstrating that while these methods perform well under MAR, they fail to generalize under certain MNAR class imbalance settings.

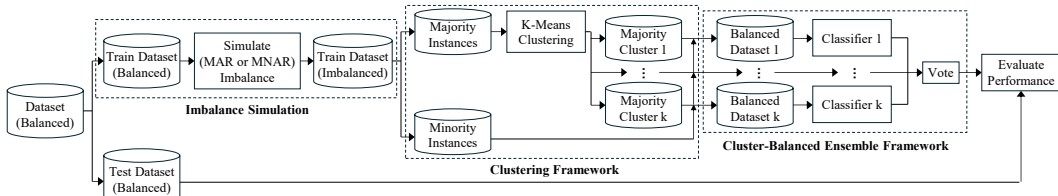

Figure 2: Overall cluster-balanced ensembling framework and evaluation protocol.

- We introduce a cluster-balanced ensembling framework built on already established components, as a first step toward addressing MNAR class imbalance.
- We characterize conditions under which MNAR most severely degrades the performance of existing techniques and most benefits from cluster-balanced ensembling.
- We introduce an evaluation protocol that uses near-balanced datasets (when available) for an unbiased assessment of MNAR imbalance correction techniques.

## 2 RELATED WORK

Imbalance techniques can be broadly categorized into data-level, cost-sensitive, and hybrid strategies (see Altalhan et al. (2025) for a recent review). Data-level methods include oversampling the minority class, such as SMOTE (Chawla et al., 2002) and its variants (Pradipta et al., 2021), which can mitigate imbalance but risk reinforcing minority class bias under MNAR. On the other hand, undersampling methods like Cluster Centroids (Yen & Lee, 2009) may lose important structure that can aid in mitigating MNAR. Cost-sensitive learning modify the weight of instances in the classification loss functions to emphasize the minority class during training (Lin et al., 2017a; Cui et al., 2019), but face similar limitations to oversampling under MNAR.

Ensemble and hybrid methods combine multiple techniques to leverage their strengths, and have gained popularity in recent years (see Khan et al. (2024) for a review). However, most ensembles rely on random undersampling, are designed mainly to reduce variance, and do not explicitly address MNAR imbalance. In contrast, our method specifically promotes diversity across the ensemble to learn informative contrasts between majority segments and minority instances to mitigate MNAR imbalance.

There also exist methods with rigorous guarantees for metrics relevant under class imbalance (e.g., F1, recall) (Diochnos & Trafalis, 2021; Narasimhan, 2018). These approaches optimize target metrics through principled reweighting, sampling, and constraint-based training (often with calibrated thresholds). Our approach is complementary to these guarantee-driven methods, and can serve as a basis by constructing near-balanced clusters under MNAR imbalance.

An important component of our proposed approach is clustering the majority class. Clustering has been used in class imbalance to undersample the majority class into a single subset (Babar & Ade, 2016; Deng et al., 2020; Lin et al., 2017b; Rayhan et al., 2017; Tsai et al., 2019; Shahabadi et al., 2021; Zhou & Sun, 2024; Hoyos-Osorio et al., 2021; Onan, 2019; Farshidvard et al., 2023; Sobhani et al., 2014), combine a single representative cluster with minority points (Rahman & Davis, 2013), or entirely replace majority points with cluster centroids (Lin et al., 2017b). Clustering has also been used to aid oversampling or reweighting the minority class (Xu et al., 2021; Singh & Dhall, 2018; Polat, 2018). However, none of these methods use clustering to segment the majority class into diverse equal-size groups with the minority instances, and combine every segment with the entire minority class to form balanced datasets to handle MNAR imbalance.

The method most similar to ours is the EKR framework by Duan et al. (2020), which uses silhouette-based clustering to identify the most homogenized majority clusters, and applies under or oversampling on the resulting clusters to generate balanced ensembles. While similar in overall methodology, our paper studies an entirely different challenge in MNAR imbalance. In particular, our method differs in key details that mitigate MNAR: we fix the number of clusters based on the imbalance ratio rather than silhouette scores to specifically generate balanced clusters. For example, under a 20:1 imbalance ratio, our method enforces 20 majority clusters, while EKR may use two, if two

clusters best homogenize the majority class (see Appendix A.3 for comparison). Furthermore, we intentionally avoid any resampling to prevent overfitting to observed minority instances and promote diversity. As we show in our experiments, these details lead to considerable gains in performance over EKR under MNAR class imbalance.

Our paper adopts the MAR and MNAR concepts from the missing data literature. While the core concept remains valid, methods in this literature typically aim to address missing feature values and not entire instances, and also do not differentiate by binary class labels. As such, they are largely inapplicable to class imbalance. The closest related work are "selection models" (e.g., based on Heckman (1979)), which correct MNAR selection bias for continuous outcomes. However, this is via strong parametric assumptions and access to external information (e.g., "exclusion variables"). In contrast, our approach handles MNAR class imbalance non-parametrically and for binary labels, without requiring a model of the selection process or access to exclusion variables.

## 3 PROPOSED METHODOLOGY

We formalize the MNAR class imbalance problem in §3.1, detail our proposed cluster-balanced ensembling approach in §3.2, and describe our proposed evaluation protocol in §3.3.

### 3.1 PROBLEM FORMALIZATION

We consider a supervised binary classification task over an *observed* dataset $\mathcal{S} = \{(X_i, y_i)\}_{i=1}^{N}$, with features $X_i = \left[x_i^1, \ldots, x_i^d\right]$ and binary labels $Y_i \in \{0, 1\}$. The observed dataset consists of $n^- = |\mathcal{S}^{\text{maj}}|$ observation from the $Y = 0$ negative or *majority* class with $\mathcal{S}^{\text{maj}} = \left\{(X_i, y_i) \in \mathcal{S}^{\text{train}} \mid y_i = 0\right\}_{i=1}^{N}$; and $n^+ = |\mathcal{S}^{\text{min}}|$ observations from the $Y = 1$, positive or *minority* class with $\mathcal{S}^{\text{min}} = \left\{(X_i, y_i) \in \mathcal{S}^{\text{train}} \mid y_i = 1\right\}_{i=1}^{N}$. In many real-world applications, $n^+ \ll n^-$ creating *class imbalance*.

The observed dataset can be considered a sample of its underlying distribution $\mathcal{S} \sim \mathcal{D}$, which is generally unknown. Let $R_i \in \{0, 1\}$ be a binary indicator that denotes whether an instance $(X_i, y_i)$ is observed from $\mathcal{D}$ with $(X_i, y_i) \in \mathcal{S}$. MAR class imbalance occurs when the observation mechanism satisfies $R_i \perp\!\!\!\perp Y_i \mid X_i$, or equivalently, $\mathbb{P}(R_i \mid X_i, Y_i) = \mathbb{P}(R \mid X_i)$. That is, under MAR, observation of minority instances are conditionally independent from their class $Y$ given the observed features $X$. On the other hand, MNAR class imbalance occurs when $R_i \not\!\perp\!\!\!\perp Y_i \mid X_i$, or equivalently, $\mathbb{P}(R_i \mid X_i, Y_i) \neq \mathbb{P}(R_i \mid X_i)$.

Under MNAR class imbalance, observations from some regions of the minority class are systematically underrepresented or even excluded from $\mathcal{S}$ with $\mathbb{P}(R_i \mid X_i, y_i = 1) < \mathbb{P}(R_i \mid X_i)$. The observed dataset thus exhibits both a marginal imbalance in $\mathbb{P}(Y_i = 1 \mid R_i = 1)$ and a distorted conditional distribution $\mathbb{P}(X_i \mid Y_i = 1, R_i = 1)$ that may differ substantially from the population-level distribution $\mathbb{P}(X_i \mid Y_i = 1)$. Note that this is different than the marginal distribution of features $\mathbb{P}(X_i)$ governed by $\mathcal{D}$. That is, an instance $(X_i, y_i)$ can be naturally less prevalent than other instance in $\mathcal{D}$, and MNAR occurs only when its probability of observation $\mathbb{P}(R_i) = \mathbb{P}((X_j, y_j) \in \mathcal{S})$ depends on its class $Y_i$ and features $X_i$.

Under MNAR, learning from $\mathcal{S}$ may not yield a model that generalizes well to the true distribution $\mathcal{D}$, presenting a fundamental challenge different to only class-count imbalance. This is particularly problematic when the goal is to understand properties of the minority class and interpret the relationship between $X_i$ and outcomes $y_i = 1$, often the very objective of class imbalance methodologies.

### 3.2 CLUSTER-BALANCED ENSEMBLING FOR MNAR CLASS IMBALANCE

Our approach to addressing MNAR class imbalance follows well-known hybrid methods of undersampling the majority class to create ensembles with observed minority instances, but with key refinements. Instead of randomly undersampling the minority instances, we use K-means clustering (with default euclidean distance) to segment the majority class $\mathcal{S}^{\text{maj}}$ into $k$ distinct subgroups $\mathbb{C} = \{C_1, \ldots, C_k\}$. We set $k = \text{round}\left(\frac{n^-}{n^+}\right)$ to increase the chance that each cluster $C_p \in \mathbb{C}$ contains approximately $n^+$ number of majority instances.

Furthermore, we perform clustering on the original unscaled dataset, even if the final classifier is trained on the normalized variant. This is to capture local structure defined by original regions of the feature space, rather than the normalized subspace that may compress important structure. In particular, normalizing before clustering often results in less diverse majority subgroups.

We next construct near-balanced datasets $\mathcal{S}_p = C_p \cup \mathcal{S}^{\mathrm{min}}$ by combining a majority cluster $C_p$ with all instances of the minority set $\mathcal{S}^{\mathrm{min}}$. Although near-balance is not guaranteed for any of the resulting datasets $\mathcal{S}_p$, its imbalance $1 - \frac{n^+}{|C_p|}$ is strictly lower than the imbalance $1 - \frac{n^+}{n^-}$ of the original dataset $\mathcal{S}$ when $k > 1$.

For each dataset $\mathcal{S}_p$, we train a base classifier $f$ to create an ensemble $\mathbb{E} = \{f_1, \ldots, f_k\}$. The base classifier can be any classification algorithm, but should remain consistent across the ensemble. This is to emphasize learning across different regions of the data, rather than maximizing local classification accuracy via different models.

Classifiers $f_p \in \mathbb{E}$ are weighted by their PR AUC performance on the overall training dataset $\mathcal{S}^{\mathrm{train}}$ (this metric provided the best generalization in our experiments, as shown in Appendix A.6), and final predictions are made by aggregating all trained classifiers according to their weight. As each classifier trains on a different conditional slice of the observed data distribution $\mathcal{D}^{\mathrm{obs}}$, its aggregation provides a unique benefit in reducing bias from underrepresented subgroups than any one cluster. The overall procedure is depicted in Figure 2, with the pseudocode given in Appendix B.

### 3.3 Balanced Gold-Standard Evaluation Protocol

A central challenge in class imbalance under MNAR is that the evaluation approach itself may be biased. That is, sampling test sets from imbalanced dataset that are potentially MNAR, can bias true model performance. To highlight and address this potential bias, we propose to measure imbalance performance using gold-standard datasets that are originally near-balanced. Given such a dataset $\mathcal{S}$, we first partition it into balanced training $\mathcal{S}^{\mathrm{train}}$ and test $\mathcal{S}^{\mathrm{test}}$ sets using stratified folds. We then simulate MAR or MNAR on $\mathcal{S}^{\mathrm{train}}$ via controlled instance deletions (based on features $X$ and class $Y = 1$ for MNAR).

All methods are applied to the resulting imbalanced training set $\mathcal{S}^{\mathrm{train}}$, but performance is evaluated on the untouched gold-standard test set. This design ensures that comparisons across methods reflect their ability to recover the predictive power lost due to MNAR imbalance, rather than their performance on the biased distribution.

It is important to note that this protocol is not meant to replace current evaluation practices, but rather to highlight the risks of benchmarking methods solely on test sets drawn from potentially MNAR datasets. As we later show in our numerical evaluations, evaluating on such biased test sets can artificially inflate or deflate performance, leading to misleading conclusions about the robustness of different methods. Our gold-standard approach instead serves as a diagnostic tool: it clarifies the extent to which algorithms can recover "true" predictive performance when imbalance is introduced.

## 4 Numerical Evaluation

Our experiments are designed to show three key findings: (1) which features trigger the most severe MNAR imbalance and degrade performance the most; (2) how existing imbalance correction methods fare versus our approach under MNAR versus MAR conditions; and (3) how standard hold-out evaluation can yield misleading results when applied to MNAR-imbalanced data.

### 4.1 Datasets

We considered both real and synthetic benchmark datasets from the PMLB repository (Olson et al., 2017) and chose ones that were near-balanced.[1] In addition to these datasets, we chose the Women Bank account dataset from Field et al. (2016) as an example of a real-world application specifically

---

[1]We note that the imbalance ratios reported in the PMLB repository were not entirely accurate; therefore, we computed the correct ratios by downloading and evaluating each dataset directly.

| Dataset | $N$ | $d$ | $n^+$ | $n^-$ | Dataset | $N$ | $d$ | $n^+$ | $n^-$ |
|---------|-----|-----|-------|-------|---------|-----|-----|-------|-------|
| WBank | 8065 | 48 | 4040 | 4025 | TwoNorm | 7400 | 20 | 3697 | 3703 |
| KR vs KP | 3196 | 36 | 1669 | 1527 | Ring | 7400 | 20 | 3736 | 3664 |
| Cancer | 569 | 30 | 212 | 357 | Chess | 3196 | 36 | 1669 | 1527 |

Table 1: Balanced datasets, their number of instances, dimension of $X$, and number of instances in the positive and negative class, and the feature used to simulate MNAR.

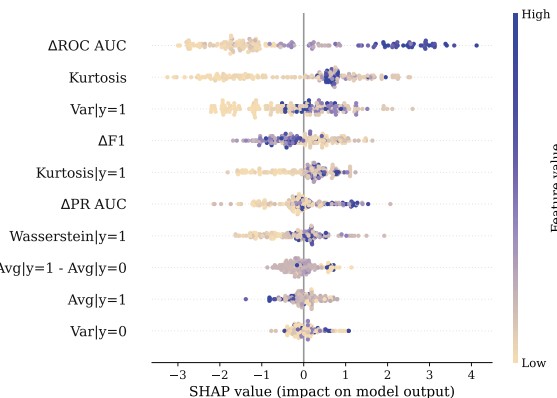

Figure 3: SHAP values of feature properties linked to poor performance of existing methods and high gains from cluster-balanced ensembling. Key factors include drop in ROC AUC, high kurtosis and variance in the positive class, low F1 drop, high PR AUC drop, large Wasserstein distance under MNAR, low mean in $y = 1$, and high mean in $y = 0$.

conducted to draw inference from the minority class. The datasets and their properties are given in Table 1.

## 4.2 FEATURES THAT LEAD TO SEVERE MNAR IMBALANCE

The impact of MNAR imbalance on existing methods varies depending on which features govern the missingness. Different features reflect different minority subpopulations, and consequently, disparately affect model learning.

In practice, MNAR mechanisms are unknown, and distinguishing MAR from MNAR often requires expert judgment. This section aims to assist experts by identifying feature types that lead to the most severe MNAR effects. In particular, we examine which features $x^j \in X$, when used to simulate MNAR imbalance, cause the most degrade of performance to existing methods are best aided by cluster-balanced ensembling.

To that end, we computed numerous properties for each feature $x^j$, including mean, variance, range, quartiles, number of potential outliers (above 0.75 quartile or below 0.25 quartile), skewness, kurtosis, number of unique values, zero fraction, average collinearity, logistic regression coefficient, random forest feature importance, permutation importance, ANOVA F-statistic, Pearson/Spearman correlation with $y$, mutual information across class, entropy, and single-variable F1. We also computed the Wasserstein distance of $x^j$, and the drop of performance in ROC AUC, PR AUC, and F1 scores between the balanced and MNAR-damaged datasets. Note that most of these metrics limit the analysis to numerical features.

These metrics were aggregated across datasets (to enable a data-set agnostic analysis) to predict which features induce MNAR that leads to poor performance by existing methods and benefit most by cluster-balanced ensembling. Our initial experiments indicated that linear models performed poorly (with at best a 65% F1 score), suggesting a complex relationship between feature properties and MNAR effects. In the end, XGBoost (with default setting and a max depth of four) trained on the ten feature properties shown in Figure 3, achieved the best predictive performance (87% F1, 94% precision, 82% recall, 90% balanced accuracy via 5-fold cross validation) .

The SHAP values from the XGBoost model (Figure 3) show that the most important properties are: drop in ROC AUC, high kurtosis and variance in the $y = 1$ class, low drop in F1, high drop in PR AUC, high Wasserstein distance between damaged and original $x$, and distinct means by class.

Overall, MNAR is most severe when imbalance is due to missing high-value minority instances that are critical for ranking instances, do not overly affect classification threshold learning, have diverse populations, and thicker tails.

### 4.3 EVALUATION FRAMEWORK

For an unbiased evaluation of imbalance methods, we construct 10-fold cross validation hold-out samples taken from the original near-balanced datasets. Then, on training data $\mathcal{S}^{\text{train}}$, we simulate 95% MAR or MNAR by deleting positive instances according to the procedure detail in Appendix B.6. The 95% threshold was used to simulate higher imbalance (we provide further results over varied MNAR deletion in Appendix A.4). Deletions were performed on the top-5 features (disclosed in Appendix B.6) that gave the highest classification probability that lead to sever MNAR according to our XGBoost model in §4.2.

We evaluate five classifiers: logistic regression (LR), support vector machines (SVM), random forests (RF), a multi-layer perceptron (MLP), and XGBoost (XGB); covering linear, kernel-based, ensemble, neural, and boosting models. Using six datasets, 10-fold cross validation, five MNAR simulations, and five base models gives 1,500 experiments for our numerical results.

To benchmark performance, we considered popular imbalance methods from four categories. (1) Cost-based: reweighting (RWGT), Focal Loss (FOC), and class-balanced loss (CLS); (2) Debiasing: Logit Adjustment (LA); (3) Ensemble-based: our proposed cluster-balanced ensemble (CBE), random-balanced ensemble (RBE) which is identical to CBE but with the distinction that the majority class is undersampled randomly, EKR, Easy Ensemble (EE), and Balanced Bagging Classifier (BB); (4) Undersampling: Cluster Centroids (CC) and Tomek Links; and (5) Oversampling: SMOTE (SMT), Borderline SMOTE (B-SMT), and ADASYN (ADSN). Implementation and details of each algorithm is given in Appendix B.1.

We also report two reference baselines. The gold standard (GLD) upper bound: performance on the original balanced datasets, and the damaged baseline (DMG) lower bound: performance on imbalanced datasets without any imbalance correction technique. These bounds are used to contextualize the gains achieved by different methods. Following prior work (Rezvani & Wang, 2023; Altalhan et al., 2025), we report F1, balanced accuracy, precision, recall, PR AUC, and ROC AUC to capture key trade-offs under MNAR class imbalance.

All algorithms are coded in Python (and given in the supplementary materials) and executed on a PC with an Intel Xeon processor W-2255 and an Ubuntu 20.04.5 operating system.

### 4.4 F1 PERFORMANCE UNDER MNAR

Table 2 gives the F1 score under 95% MNAR imbalance. All classifiers experience notable performance drops when trained on damaged data (DMG column), confirming the adverse effect of MNAR. Our proposed CBE consistently outperforms all other methods, followed by EKR and CC, which are also cluster-based techniques. CBE achieves an average 25.3% improvement in F1 across all approaches, ranging from 9% (over EKR) to 52% (over LA), and narrowly surpassed in only two settings—TwoNorm and Cancer datasets with MLP classifier, by margins of 0.8% and 2.9%, respectively.

CBE yields especially large gains over RBE, with an average F1 increase of 17.25%. This highlights that clustering the majority subpopulations is more effective in preserving decision boundaries under MNAR imbalance than random sampling.

Ensemble-based methods perform best under MNAR, while cost-sensitive and oversampling techniques perform worst. Despite their popularity, these approaches can reinforce existing minority class bias under MNAR—so much so that, in some cases, models trained on reweighted or oversampled data underperform even the damaged LB. This underscores the risk of applying imbalance correction without considering the missingness mechanism.

| $f$ | Dataset | UB | Cost-based | | | Debias | Ensemble-based | | | | | Under-sample | | Over-sample | | | LB |
|---|---|---|---|---|---|---|---|---|---|---|---|---|---|---|---|---|---|
| | | GLD | RWGT | FOC | CLS | LA | RBE | CBE | EKR | EE | BB | CC | TMK | SMT | B-SMT | ADSN | DMG |
| LR | WBank | 84.4 | 52.2 | 44.7 | 47.5 | 2.6 | 52.6 | **73.2** | 59.4 | 52.7 | 50.4 | 72.5 | 25.0 | 32.7 | 32.2 | 32.3 | 23.8 |
| | KR vs KP | 96.0 | 62.0 | 67.6 | 54.9 | 11.8 | 53.9 | **74.8** | 71.5 | 54.1 | 53.0 | 58.3 | 45.1 | 59.5 | 55.6 | 61.0 | 45.2 |
| | Cancer | 95.8 | 59.7 | 37.2 | 59.2 | 48.1 | 60.6 | **62.4** | 57.0 | 59.1 | 57.1 | 54.9 | 56.0 | 58.7 | 59.2 | 58.5 | 56.0 |
| | TwoNorm | 97.8 | 91.4 | 6.0 | 93.3 | 59.5 | 95.6 | **96.4** | 95.7 | 95.7 | 95.7 | 96.4 | 85.8 | 89.7 | 90.3 | 90.4 | 85.5 |
| | Ring | 77.0 | 42.8 | 66.5 | 36.0 | 0.0 | 48.0 | **76.2** | 68.1 | 47.9 | 47.0 | 52.1 | 0.3 | 40.5 | 13.2 | 44.6 | 0.2 |
| | Chess | 96.4 | 64.1 | 67.4 | 55.6 | 12.1 | 54.5 | **74.1** | 73.0 | 54.6 | 53.1 | 60.8 | 45.7 | 61.4 | 58.3 | 63.0 | 45.6 |
| | **Avg** | 91.2 | 62.0 | 48.2 | 57.8 | 22.3 | 60.9 | **76.2** | 70.8 | 60.7 | 59.4 | 65.8 | 43.0 | 57.1 | 51.5 | 58.3 | 42.7 |
| SVM | WBank | 83.7 | 19.6 | 0.4 | 18.8 | 6.7 | 37.4 | **67.2** | 52.1 | 37.3 | 34.7 | 42.2 | 23.9 | 20.1 | 21.2 | 19.4 | 22.5 |
| | KR vs KP | 98.3 | 47.8 | 0.0 | 40.5 | 42.9 | 43.4 | **70.8** | 61.4 | 43.3 | 42.1 | 43.4 | 55.6 | 46.2 | 45.8 | 45.9 | 55.7 |
| | Cancer | 96.1 | 60.0 | 8.0 | 56.0 | 26.3 | 77.4 | **86.4** | 78.6 | 77.9 | 73.5 | 76.5 | 60.6 | 60.9 | 60.3 | 58.5 | 60.6 |
| | TwoNorm | 97.8 | 78.5 | 0.0 | 79.1 | 53.5 | 95.4 | **96.2** | 93.0 | 91.6 | 89.6 | 92.0 | 79.0 | 70.9 | 74.3 | 71.4 | 78.8 |
| | Ring | 97.9 | 71.3 | 0.4 | 69.2 | 28.6 | 91.1 | **95.3** | 91.5 | 91.2 | 89.1 | 61.8 | 69.3 | 67.8 | 71.8 | 69.2 | 68.5 |
| | Chess | 98.6 | 50.2 | 0.0 | 40.6 | 43.2 | 43.7 | **69.9** | 61.5 | 43.7 | 42.2 | 43.6 | 56.8 | 47.9 | 46.6 | 47.9 | 57.1 |
| | **Avg** | 95.4 | 54.6 | 1.5 | 50.7 | 33.5 | 64.1 | **81.0** | 73.0 | 64.2 | 61.9 | 59.9 | 57.5 | 52.3 | 53.3 | 52.1 | 57.2 |
| RF | WBank | 80.3 | 15.4 | 11.5 | 15.3 | 0.2 | 30.8 | **73.8** | 38.8 | 31.0 | 29.4 | 63.6 | 17.0 | 19.6 | 18.9 | 19.3 | 15.7 |
| | KR vs KP | 99.1 | 41.2 | 51.4 | 41.4 | 7.0 | 43.8 | **74.4** | 59.6 | 44.3 | 43.1 | 50.1 | 43.2 | 44.6 | 42.1 | 44.6 | 43.4 |
| | Cancer | 93.8 | 50.3 | 61.2 | 40.9 | 11.5 | 74.2 | **83.5** | 76.4 | 73.4 | 72.0 | 75.7 | 55.6 | 54.6 | 53.1 | 52.4 | 55.6 |
| | TwoNorm | 97.1 | 24.5 | 34.0 | 23.8 | 0.1 | 52.0 | **80.8** | 58.4 | 50.0 | 45.0 | 76.9 | 21.6 | 33.4 | 32.3 | 35.4 | 21.6 |
| | Ring | 95.3 | 26.0 | 46.1 | 24.2 | 0.2 | 36.1 | **56.9** | 36.7 | 36.2 | 34.8 | 31.5 | 11.9 | 27.0 | 10.2 | 27.0 | 11.8 |
| | Chess | 99.1 | 40.8 | 50.0 | 41.1 | 6.0 | 44.5 | **73.3** | 66.0 | 45.0 | 43.3 | 51.6 | 43.4 | 45.5 | 41.6 | 45.2 | 43.6 |
| | **Avg** | 94.1 | 33.0 | 42.4 | 32.6 | 4.2 | 46.9 | **73.8** | 56.0 | 46.6 | 44.6 | 58.3 | 32.1 | 37.4 | 33.0 | 37.3 | 32.0 |
| MLP | WBank | 79.4 | 28.0 | 26.5 | 27.9 | 17.7 | 54.5 | **70.1** | 68.8 | 54.9 | 50.1 | 63.4 | 26.4 | 26.7 | 26.2 | 26.7 | 25.2 |
| | KR vs KP | 99.4 | 52.9 | 52.0 | 52.8 | 33.3 | 56.1 | **70.9** | 66.8 | 59.0 | 56.4 | 56.3 | 44.4 | 51.9 | 51.3 | 52.1 | 44.3 |
| | Cancer | 96.4 | 81.1 | 79.2 | 81.1 | 16.8 | 89.9 | **90.9** | 88.0 | 89.5 | 89.0 | 90.9 | 71.3 | 80.3 | 79.0 | 76.3 | 71.3 |
| | TwoNorm | 96.9 | 87.6 | 85.6 | 87.7 | 67.6 | 95.4 | 95.8 | 95.6 | **96.5** | 96.4 | 96.1 | 85.0 | 87.1 | 87.2 | 87.0 | 84.7 |
| | Ring | 93.2 | 54.5 | 53.9 | 55.0 | 30.7 | 76.7 | **78.2** | 77.4 | 73.1 | 69.7 | 59.2 | 49.7 | 53.8 | 52.1 | 53.4 | 49.5 |
| | Chess | 99.4 | 54.2 | 54.1 | 54.2 | 34.5 | 56.7 | **71.4** | 66.4 | 59.7 | 56.8 | 57.5 | 45.9 | 53.1 | 51.7 | 53.5 | 46.0 |
| | **Avg** | 94.1 | 59.7 | 58.5 | 59.8 | 33.4 | 71.5 | **79.1** | 77.4 | 72.0 | 69.7 | 70.6 | 53.8 | 58.8 | 57.9 | 58.1 | 53.5 |
| XGB | WBank | 81.2 | 20.3 | 0.0 | 25.9 | 11.2 | 36.3 | **72.8** | 41.8 | 36.5 | 31.5 | 52.8 | 21.2 | 23.3 | 23.2 | 23.4 | 20.3 |
| | KR vs KP | 99.5 | 46.2 | 48.1 | 45.9 | 35.4 | 45.0 | **65.6** | 59.3 | 44.9 | 42.9 | 49.0 | 46.2 | 53.1 | 44.4 | 54.7 | 46.2 |
| | Cancer | 95.1 | 44.5 | 53.0 | 40.7 | 0.0 | 54.8 | **55.7** | 50.5 | 53.1 | 54.6 | 54.5 | 44.5 | 40.5 | 40.9 | 43.2 | 44.5 |
| | TwoNorm | 96.9 | 32.1 | 0.0 | 39.6 | 20.4 | 41.7 | **53.6** | 43.3 | 41.7 | 37.2 | 49.2 | 32.2 | 38.4 | 37.9 | 40.7 | 32.1 |
| | Ring | 96.7 | 24.1 | 2.7 | 30.2 | 11.5 | 34.3 | **60.6** | 36.1 | 34.1 | 31.6 | 29.9 | 24.4 | 32.4 | 26.0 | 33.0 | 24.1 |
| | Chess | 99.5 | 48.7 | 49.2 | 48.0 | 35.7 | 47.3 | **68.8** | 66.8 | 47.4 | 43.5 | 51.3 | 48.9 | 56.2 | 46.7 | 58.6 | 48.7 |
| | **Avg** | 94.8 | 36.0 | 25.5 | 38.4 | 19.0 | 43.2 | **62.9** | 49.7 | 43.0 | 40.2 | 47.8 | 36.2 | 40.7 | 36.4 | 42.3 | 36.0 |
| **Overall Average** | | 93.94 | 49.07 | 35.22 | 47.85 | 22.51 | 57.33 | **74.58** | 65.37 | 57.30 | 55.15 | 60.48 | 44.53 | 49.26 | 46.43 | 49.61 | 44.27 |

Table 2: F1 scores with MNAR imbalance. Our proposed CBE achieved highest average performance.

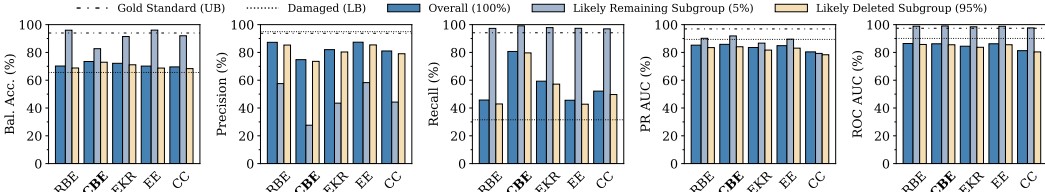

Figure 4: Average performance under MNAR imbalance, across classifiers and datasets for top-performing F1 methods. The main advantage of CBE is higher recall but lower precision. The higher precision and AUC performance of competing methods at lower recall shows overfitting to observed minority instances, without generalizing to the entire minority population.

## 4.5 BALANCED ACCURACY, PRECISION, RECALL, PR AUC, AND ROC AUC PERFORMANCE UNDER MNAR

Figure 4 gives the balanced accuracy, precision, recall, PR AUC, and ROC AUC of the most promising MNAR imbalance techniques, averaged over all datasets and classifiers. Results are reported over the overall test set $\mathcal{S}^{\text{test}}$, but also subsets of $\mathcal{S}^{\text{test}}$ that are likely to be MNAR according to the deletion procedure applied to $\mathcal{S}^{\text{train}}$. This gives approximate results on how methods perform on test instances similar to the ones observed in the training data, versus ones that are MNAR.

The major advantage of CBE is higher recall, with an increase of approximately 25% over the next best approach. This indicates that CBE can better generalize and detect unobserved minority

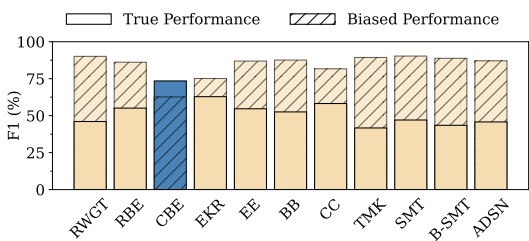

Figure 5: F1 scores from hold-out sets sampled from MNAR-damaged data, averaged across classifiers and datasets. Ignoring the imbalance source leads to misleading F1 scores that fail to reflect true performance on the underlying minority distribution.

instances under MNAR, albeit at approximately 12% lower precision. Subgroup results confirm this, with CBE significantly outperforming other methods in detection instances likely to be MNAR.

CBE also shows approximately 1.5–2% higher balanced accuracy and AUC measures over competing methods. However, evaluating MNAR imbalance performance using precision or AUC measures alone gives misleading results. For instance, the damaged LB baseline deceptively outperforms all methods in precision and both PR and ROC AUC, but at considerably lower recall. This indicates strong overfitting to the few observed minority instances, and poor generalization to the unobserved minority population. As such, evaluating MNAR imbalance using metrics that disproportionately reward high precision at the cost of recall may misleadingly claim superiority over more generalizable models.

### 4.6 BIASED EVALUATION BY SAMPLING MNAR DATA

Standard evaluation protocols often overstate model performance by sampling both training and test sets from the same MNAR-damaged data. To reveal this flaw, we simulate 85% MNAR deletion (to retain sufficient instances for subsequent hold-out samples), and evaluate models on hold-out sets drawn solely from the damaged dataset (mimicking conventional practice) to end at 95% MNAR. This setup mirrors real-world conditions where the test data shares the same structural bias as training, but it fails to reflect the true minority population.

The resulting average F1 scores across all datasets and classifiers are shown in Figure 5. Under MNAR imbalance, evaluating on data drawn from the same biased distribution that models train on can overstate performance and obscure generalization. In particular, this biased evaluation misleadingly favors models that overfit to the observed minority subset while penalizing those that generalize to the entire population. These results reinforce the importance of accounting for the nature of imbalance when evaluating mitigation methods.

## 5 CONCLUSION

This paper studies MNAR class imbalance, where missingness depends jointly on class labels and predictive features. Under MNAR, popular imbalance methods fail to generalize and instead amplify biases in the observed data.

To address this, we construct an ensemble of cluster-balanced datasets that span diverse, distinct regions of the majority class, and contrast them to minority points. Our method outperforms state-of-the-art techniques under MNAR by an average increase of 25% in F1 score, when missingness is driven by highly informative features. We also introduce an evaluation protocol that reveals how standard validation practices can misrepresent performance under MNAR.

Our findings emphasize that understanding why imbalance occurs is as important as how much imbalance exists. Hence, we shift the focus from imbalance in class counts to the missingness mechanism in class imbalance.

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

## A  ADDITIONAL EVALUATIONS

Section A.1 compared F1 performance of competing methods under MAR. We tune to F1 performance in §A.2 to assess whether established methods can be adapted to identify MNAR instances by adjusting the classification threshold. In §A.3 we report distinct statistics on the underlying clustering mechanisms used in CBE, RBE, and EKR. We perform sensitivity analyses and ablation studies, by varying MNAR class imbalance in §A.4, varying the number of clusters in CBE in §A.5, using alternative metrics for our CBE voting mechanism in §A.6.

### A.1  F1 PERFORMANCE UNDER MAR

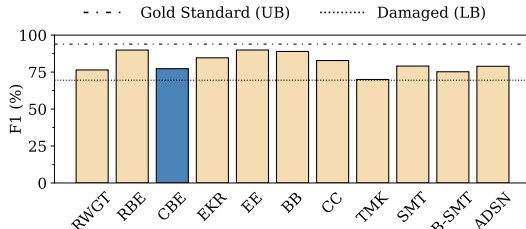

Figure 6: F1 scores with MAR imbalance. Ensembling approaches perform best under MAR imbalance.

Figure 6 gives the average performance of all methods under MAR imbalance. Since the minority class retains sufficient representation under MAR, existing methods excel in mitigating imbalance. In particular, ensemble methods like EE, RBE, and BB, and also CC undersampling perform best, while, CBE, reweighting, and oversampling perform worse. This shows that CBE's reliance on majority clusters for balancing can inject irrelevant or redundant information when majority clusters are not associated to missingness.

### A.2  THRESHOLD TUNING FOR F1 PERFORMANCE

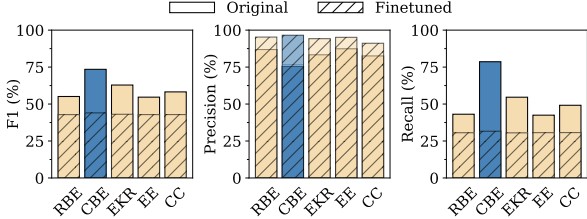

Figure 7: Threshold tuning algorithms to maximize F1 score. Tuning leads to higher precision but considerably lower recall, overfitting the algorithms to observed minority instances.

We next tuned the classification threshold of all algorithms to maximize F1 score on the training set. As Figure 7 shows, this has a consistent negative effect, and leads to overfitting the observed minority instances. All algorithms, and in particular CBE, show an increase in precision, but a large drop in recall. As such, they are very accurate in predicting the observed minority instances, but lose generalization to the entire minority class population.

### A.3 NUMBER OF CLUSTERS, PER-CLUSTER IMBALANCE, SILHOUETTE SCORE

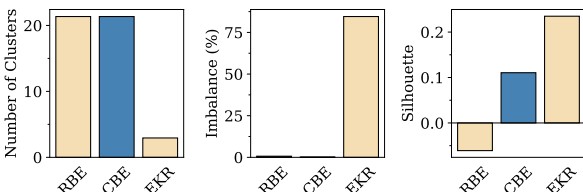

Figure 8: Number of clusters, average per-cluster imbalance, and silhouette distance of cluster-based approaches.

All three approaches of RBE, CBE, and EKR involve clustering the majority class into $k$ cluster $\mathbb{C} = \{C_1, \ldots, C_k\}$, and combining each cluster $C_p \in \mathbb{C}$ with the entire minority instances $\mathcal{S}^{\min}$ to create datasets $\{\mathcal{S}_1, \ldots, \mathcal{S}_k\}$. While the goal of clustering in CBE and RBE is to create near-balanced datasets $\mathcal{S}_p$, the goal of EKR is to create the most homogenized majority clusters $C_p$. Figure 8 demonstrates the difference of these approaches in the number of clusters created, the average percentage of imbalance $1 - \frac{n^+}{|C_p|}$ between clusters $C_p$ and all $n^+$ minority instances, and the average Silhouette score measuring the homogeneity of each cluster $C_p$, over all datasets. We show imbalance for EKR prior to its over-sampling step (where each cluster is oversampled to address imbalance) to narrow the analysis on only the clustering step.

Both CBE and RBE generate almost 10 times the number of clusters compared to EKR. This leads to almost zero imbalance in the resulting datasets $\mathcal{S}_p$ for both methods, while imbalance remains high after EKR clustering. By design, the most homogenized clusters are produce by EKR, followed by CBE. As expected RBE shows negative silhouette as subsets are chosen randomly. These results highlight the fundamental differences between the tree methods, despite simillarities in overall approach.

### A.4 VARYING MNAR CLASS IMBALANCE

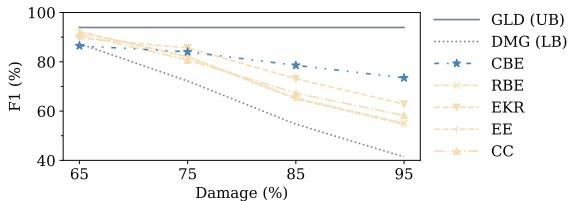

Figure 9: F1 score under varying MNAR degree, averaged over all classifiers and datasets. The advantage of CBE grows with larger MNAR imbalance.

Figure 9 shows how F1 score changes with increasing MNAR imbalance for the top-performing methods, averaged across datasets and classifiers. CBE performs best at higher MNAR levels (beyond 80%), with its advantage growing as imbalance increases. This is because lower imbalance thresholds retain more of the underrepresented minority instances, and are closer to MAR imbalance scenarios.

## A.5 VARYING NUMBER OF CLUSTERS

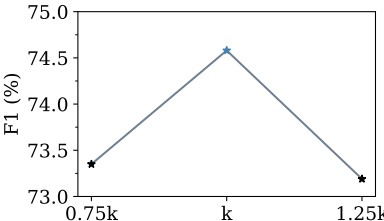

Figure 10: F1 score under varying $k$, averaged over all classifiers and datasets. The highest performance is achieved when $k = \text{round}\left(\frac{n^-}{n^+}\right)$.

Our CBE approach is designed to create $k = \text{round}\left(\frac{n^-}{n^+}\right)$ clusters of majority points, with the goal of achieving balanced datasets $\mathcal{S}_p$. Figure 10 shows the effects of varying $k$ on the F1 score. The best performance is achieved with the original setting, with both increasing and decreasing $k$ lowering the F1 score.

## A.6 ALTERNATIVE VOTING METRICS

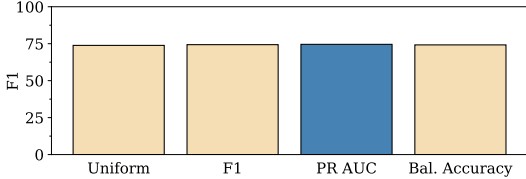

Figure 11: F1 score under different voting metrics for CBE, averaged over all classifiers and datasets. The highest performance is achieved when the ensemble votes based on PR AUC.

Figure 11 gives the average F1 score under different voting metrics of the ensemble $\{f_1, \ldots, f_2\}$. The metrics include the original used PR AUC, F1 score, balanced accuracy, and uniform voting (where each $f_i$ is given a constant 1 vote). Results remain relatively stable over different voting metrics, with PR AUC providing small (less than 1%) advantage over the other metrics).

# B ALGORITHMS

A brief introduction of competing methods and their hyperparameters is given in §B.1, followed by our overall MAR and MNAR simulation procedure in §B.6. The pseudocodes of our proposed CBE, and random undersampling ensemble counterpart RBE is given in §B.7.

## B.1 COMPETING METHODS AND HYPERPARAMETERS

To benchmark performance, we considered popular class-imbalance methods spanning cost-based reweighting, debiasing, ensemble learning, and data resampling. For fairness, we used standard implementations (scikit-learn, imbalanced-learn, or PyTorch) and widely adopted default hyperparameters, tuning only where explicitly noted. The implementation code of all algorithms is provided in the supplementary materials.

### B.1.1 COST-BASED

Reweighting (RWGT): Each minority instance in $\mathcal{S}^{\min}$ is weighted inversely to its frequency in the training set. In particular, the minority class weight is set to $\frac{N}{2n^-}$ and the majority to $\frac{N}{2n^+}$.

Focal Loss (FOC): Extends cross-entropy by down-weighting easy examples (Lin et al., 2017a). The loss for a true class with predicted probability $\mathbb{P}_t = -(1 - \mathbb{P}_t)^\gamma \log(\mathbb{P}_t)$. We use the canonical defaults $\alpha = 0.25$ (minority class weight) and $\gamma = 2$ (focusing parameter).

Class-Balanced Loss (CLS): Reweights classes using their "effective number of samples" $E(n_y) = (1 - \beta^{n_y})/(1 - \beta)$ (Cui et al., 2019). Class weight is proportional to $\frac{1}{E(n_y)}$. Following Cui et al. (2019), we set $\beta = \frac{(n-1)}{n}$, the standard default.

## B.2 DEBIASING

Logit Adjustment (LA): Corrects for skewed class priors by adding $\tau \log \pi_y$ to each class logit before cross-entropy, where $\pi_y$ is the empirical class prior (Menon et al., 2020). We use the standard setting $\tau = 1$.

## B.3 ENSEMBLE-BASED METHODS

Random-Balanced Ensemble (RBE): Identical to CBE but replaces clustering with random under-sampling of the majority to form balanced subsets. Further details are provided in §B.7.

EKR: Follows Duan et al. (2020), where the majority class is partitioned by K-means and one representative per cluster is used with all minority instances in each base learner.

Easy Ensemble (EE): Constructs multiple balanced bootstrap samples of the dataset and trains AdaBoost learners on each set (Liu et al., 2008). We use the `EasyEnsembleClassifier` Python library from imbalanced-learn with default hyperparameters.

Balanced Bagging (BB): Implements bagging with balanced bootstrap samples. We use the `BalancedBaggingClassifier` Python library from imbalanced-learn (LemaÃŽtre et al., 2017) with default hyperparameters.

## B.4 UNDERSAMPLING METHODS

Cluster Centroids (CC): Replaces the majority class with its $k$ cluster centroids, reducing its size to match the minority count (Yen & Lee, 2009). We use the `ClusterCentroids` Python library from imbalanced-learn (LemaÃŽtre et al., 2017) with default hyperparameters.

Tomek Links: Undersamples the majority set by removing borderline instances that form Tomek pairs (nearest neighbors of opposite class). We use the `TomekLinks` Python library from imbalanced-learn (LemaÃŽtre et al., 2017) with default hyperparameters.

## B.5 OVERSAMPLING METHODS

SMOTE (SMT): Generates synthetic minority samples by interpolating between each minority point and its nearest neighbors (Chawla et al., 2002). We use the `SMOTE` Python library from imbalanced-learn (LemaÃŽtre et al., 2017) with default hyperparameters.

Borderline SMOTE (B-SMT): A variant of SMOTE that oversamples only minority instances near the decision boundary (with many majority neighbors) (Han et al., 2005). We use the `BorderlineSMOTE` Python library from imbalanced-learn (LemaÃŽtre et al., 2017) with default hyperparameters.

Adaptive synthetic oversampling (ADSN): Generates more synthetic points in regions where the minority is sparsely represented relative to the majority (He et al., 2008). We use the `ADASYN` Python library from imbalanced-learn (LemaÃŽtre et al., 2017) with default hyperparameters.

## B.6 MAR AND MNAR SIMULATION

Algorithm 1 gives the pseudocode of our damaging process to simulate MAR or MNAR class imbalance from a balanced dataset. The damaging procedure simulates MNAR imbalance via a soft deletion mechanism that incrementally removes minority instances based on higher or lower values

| Dataset | Features used to simulate MNAR | | | | |
|---|---|---|---|---|---|
| | $x^1$ | $x^2$ | $x^3$ | $x^4$ | $x^5$ |
| WBank | T1wFinlit | T2wFinlit | cont cl age female | GPcont c ghost | GPcont c sarpanch sc |
| KR vs KP | reskr | thrsk | skach | hdchk | rimmx |
| Cancer | area worst | perimeter worst | area se | compactness worst | radius se |
| TwoNorm | A13 | A18 | A6 | A1 | A20 |
| Ring | A1 | A13 | A14 | A5 | A12 |
| Chess | A20 | A30 | A25 | A14 | A21 |

Table 3: Datasets and the top-5 features used to simulate MNAR.

in a target feature $X^{\text{dmg}}$. Features $X^{\text{dmg}}$ are given in Table 3, and are chosen based on the top-5 results of the XGBoost model detailed in §4.2, indicating severe MNAR damage. The direction of deletion (higher or lower values) is determined by the logistic regression coefficient $\beta$ sign of $X^{\text{dmg}}$. If $\beta > 0$, then we delete instances in ascending, and otherwise, in descending order of $X^{\text{dmg}}$.

Instead of deterministically deleting entire regions of the feature space, deletion likelihoods vary smoothly with the ordering variable $X^{\text{dmg}}$, ensuring that some minority points remain observable across all regions. This preserves partial support of the original distribution $\mathcal{D}$ while inducing systematic distortions in its tail regions. Such a design captures realistic MNAR settings, where missingness arises from bias that skews but does not fully censor subpopulations. By applying deletions progressively at increasing thresholds (65%, 75%, 85%, 95%), we generate a controlled family of datasets that enable robust and comparable evaluation of methods under varying MNAR severity.

---

**Algorithm 1** MAR and MNAR Damaging Process

**Require:** Train dataset $\mathcal{S}^{\text{train}} = \left\{\mathcal{S}^{\text{maj}}, \mathcal{S}^{\text{min}}\right\}$, desired damage level $\pi^{\text{tar}}$, damaging feature $X^{\text{dmg}}$, damage type how $\in \{\text{MAR}, \text{MNAR}\}$
1: Initialize set of damaged indices $\mathcal{I}^{\text{dam}} \leftarrow \emptyset$
2: Copy dataset $\bar{\mathcal{S}}^{\text{train}} \leftarrow \mathcal{S}^{\text{train}}$,
3: **for** $\pi \in \{0.65, 0.75, 0.85, 0.95\}$ **do**
4:      Reset dataset $\bar{\mathcal{S}}^{\text{train}} \leftarrow \mathcal{S}^{\text{train}}$
5:      Set $n^{\text{tar}} \leftarrow \pi \times \left|\mathcal{S}^{\text{min}}\right|$
6:      **if** how $=$ rnd **then**
7:          Sample $n^{\text{tar}}$ points at random: $\mathcal{I}^{\text{dam}} \leftarrow \mathcal{I}^{\text{dam}} \cup \text{Sample}\left(\{i \in \bar{\mathcal{S}}^{\text{train}} : y_i = 1\}, n^{\text{tar}}\right)$
8:      **else**
9:          Sort $\bar{\mathcal{S}}^{\text{train}}$ by feature $X^{\text{dmg}}$
10:          **while** $\left|\mathcal{I}^{\text{dam}}\right| < n^{\text{tar}}$ **do**
11:              $\theta \leftarrow 0.9 - \dfrac{0.1 \times \left|\mathcal{I}^{\text{dam}}\right|}{n^{\text{tar}}}$
12:              **for** $i \in \bar{\mathcal{S}}^{\text{train}}$ **do**
13:                  **if** $y_i = 1$ **and** $\text{Uniform}(0, 1) \leq \theta$ **then**
14:                      $\mathcal{I}^{\text{dam}} \leftarrow \mathcal{I}^{\text{dam}} \cup \{i\}$
15:                      $\theta \leftarrow \theta - 0.1/n^{\text{tar}}$
16:                  **end if**
17:                  **if** $\left|\mathcal{I}^{\text{dam}}\right| = n^{\text{tar}}$ **then**
18:                      **break**
19:                  **end if**
20:              **end for**
21:          **end while**
22:      **end if**
23:      **if** $\pi = \pi^{\text{tar}}$ **then**
24:          **break**
25:      **end if**
26: **end for**
27: **return** $\mathcal{S}^{\text{train}} \setminus \mathcal{I}^{\text{dam}}$

---

### B.7 CBE AND RBE PSEUDOCODES

Algorithm 2 gives the pseudocode for the RBE approach.

---

**Algorithm 2** Cluster-Balanced Ensemble (CBE)

---

**Require:** Training dataset $\mathcal{S}^{\text{train}} = \left\{ \mathcal{S}^{\text{maj}}, \mathcal{S}^{\text{min}} \right\}$, Base classifier $f$, Test dataset $\mathcal{S}^{\text{test}}$

1: $k \leftarrow \text{round} \left( \frac{|\mathcal{S}^{\text{maj}}|}{|\mathcal{S}^{\text{min}}|} \right)$
2: Initialize sum of votes $V \leftarrow 0$
3: Initialize probability matrices: $P^{\text{train}} \leftarrow \mathbf{0}_{|y^{\text{train}}|}, P^{\text{test}} \leftarrow \mathbf{0}_{|y^{\text{test}}|}$
4: **for** $r = 1 \rightarrow k$ **do**
5:     Apply K-means clustering on $\mathcal{S}^{\text{maj}}$ with $k$ cluster centers and 10 random initializations
6:     **for** $p = 1 \rightarrow k$ **do**
7:         $\mathcal{S}_p \leftarrow C_p \cup \mathcal{S}^{\text{min}}$
8:         $\hat{f} \leftarrow \text{Fit } f \text{ on } \mathcal{S}_p$
9:         Predict probabilities: $\hat{P}^{\text{train}} \leftarrow \hat{f}(X^{\text{train}}), \hat{P}^{\text{test}} \leftarrow \hat{f}(X^{\text{test}})$
10:        Compute vote $v \leftarrow \text{PR AUC} \left( y^{\text{train}}, \quad \hat{P}^{\text{train}} \right)$
11:        Update $V \leftarrow V + v, P^{\text{train}} \leftarrow P^{\text{train}} + \hat{P}^{\text{train}} \times v, P^{\text{test}} \leftarrow P^{\text{test}} + \hat{P}^{\text{test}} \times v$
12:     **end for**
13: **end for**
14: Normalize: $P^{\text{train}} \leftarrow \frac{P^{\text{train}}}{V}, \quad P^{\text{test}} \leftarrow \frac{P^{\text{test}}}{V}$
15: **return** $P_{\text{train}}, P_{\text{test}}$

---

Algorithm 3 gives the pseudocode for the RBE approach.

---

**Algorithm 3** Random-Balanced Ensemble (RBE)

---

**Require:** Training dataset $\mathcal{S}^{\text{train}} = \left\{ \mathcal{S}^{\text{maj}}, \mathcal{S}^{\text{min}} \right\}$, Base classifier $f$, Test dataset $\mathcal{S}^{\text{test}}$

1: $k \leftarrow \text{round} \left( \frac{|\mathcal{S}^{\text{maj}}|}{|\mathcal{S}^{\text{min}}|} \right)$
2: Initialize sum of votes $V \leftarrow 0$
3: Initialize probability matrices: $P^{\text{train}} \leftarrow \mathbf{0}_{|y^{\text{train}}|}, P^{\text{test}} \leftarrow \mathbf{0}_{|y^{\text{test}}|}$
4: $\tilde{\mathcal{S}}^{\text{maj}} = \mathcal{S}^{\text{maj}}$
5: **for** $r = 1 \rightarrow k$ **do**
6:     $\bar{\mathcal{S}}^{\text{maj}} \leftarrow$ Randomly undersample $\min \left\{ \left| \tilde{\mathcal{S}}^{\text{maj}} \right|, \left| \mathcal{S}^{\text{min}} \right| \right\}$ instances from $\tilde{\mathcal{S}}^{\text{maj}}$ without replacement
7:     Update $\tilde{\mathcal{S}}^{\text{maj}} \leftarrow \tilde{\mathcal{S}}^{\text{maj}} \setminus \bar{\mathcal{S}}^{\text{maj}}$
8:     $\mathcal{S}_p \leftarrow \bar{\mathcal{S}}^{\text{maj}} \cup \mathcal{S}^{\text{min}}$
9:     $\hat{f} \leftarrow \text{Fit } f \text{ on } \mathcal{S}_p$
10:    Predict probabilities: $\hat{P}^{\text{train}} \leftarrow \hat{f}(X^{\text{train}}), \hat{P}^{\text{test}} \leftarrow \hat{f}(X^{\text{test}})$
11:    Update $P^{\text{train}} \leftarrow P^{\text{train}} + \hat{P}^{\text{train}}, P^{\text{test}} \leftarrow P^{\text{test}} + \hat{P}^{\text{test}}$
12: **end for**
13: Normalize: $P^{\text{train}} \leftarrow \frac{P^{\text{train}}}{k}, \quad P^{\text{test}} \leftarrow \frac{P^{\text{test}}}{k}$
14: **return** $P_{\text{train}}, P_{\text{test}}$

---

## C THE USE OF LARGE LANGUAGE MODELS

We used large language models (LLMs) solely to polish the writing of this paper. All ideas, methods, analyses, and results were developed by the authors, with LLM assistance limited to improving clarity and readability of the text.

