# OpenReview forum: "Not All Imbalance Is Random: Cluster-Balanced Ensembling for Missing-Not-At-Random Class Imbalance"
_ICLR.cc/2026/Conference — Submitted to ICLR 2026_

### Official Review · Reviewer_jUqk · 2025-10-26

**Soundness:** 3
**Presentation:** 2
**Contribution:** 3
**Rating:** 6
**Confidence:** 5

**Summary:**

In this paper, the authors study the problem of class imbalance. To be specific, the authors argue that the test distribution is often imbalanced not in a random way: Data is missing-not-at-random (MNAR), which is not taken into account in the class imbalance literature so far. The authors provide an analysis and introduce a method borrowed from the MNAR literature. The introduced ensembling based method provides strong results on two tabular benchmarks.

**Strengths:**

+ MNAR appears to be an important issue overlooked in the class imbalance literature.
+ The provided analysis is helpful.
+ The introduced method borrowed from the MNAR literature appears to be effective.

**Weaknesses:**

Weaknesses:

1. The paper makes many claims without providing justifying references. The whole Introduction section is written without any references at all.

2. They generate MAR and MNAR from a balanced dataset. Although this is understandable for an analysis, it is not clear what the scale of the problem is for naturally-collected long-tailed datasets. Therefore, it is not clear whether this is a real concern in natural datasets. I strongly suggest the authors to perform experiment with LT datasets such as ImageNet-LT, Places-LT, iNaturalist.

2.1. The authors argue that "sampling test sets from imbalanced dataset that are potentially MNAR, can bias true model performance." => This potential should not prevent one from performing experiments with such datasets.

3. The figures in the paper have major issues.

3.1. Figure 1 is not referred to in the text. As the figure doesn't explain how the samples are generated or what kind of dataset these are, the figure fails to support the paper.

3.2. Fig 2: Text too small to read.

3.3. Figure 3 is very critical for the main motivations of the paper. However, (i) many variables (measures, SHAP values, feature values) are drawn in a complex manner without sufficient guidance in the figure or the caption, and (ii) the figure is not sufficiently explained in the text.

4. I find the experimental evaluation weak.

4.1. For starters, the datasets are not used in the imbalance literature and therefore, the experimental evaluation fails to be convincing. Even for a balanced starting dataset, the paper could have preferred datasets commonly used: E.g., CIFAR10 and CIFAR100, which are converted into their balanced settings in a controlled manner.

4.2. "These metrics were aggregated across datasets (to enable a data-set agnostic analysis) to predict which features induce MNAR that leads to poor performance by existing methods and benefit most by cluster-balanced ensembling." => Is the impact of the missingness of a feature not dataset-dependent? Aggregation over datasets makes it difficult to perform a problem-dependent deductions.



Minor comments:

- "follows well-known hybrid methods of undersampling the majority class" => Please cite.
- "euclidean distance" => "Euclidean distance".
- "This is to capture local structure defined by original regions of the feature space, rather than the normalized subspace that may compress important structure. In particular, normalizing before clustering often results in less diverse majority subgroups." => It would be nice to visualize this.
- "sever MNA" => "severe MNA".
- "Implementation and details of each algorithm is given" => "Implementation and details of each algorithm are given".

**Questions:**

Please see Weaknesses.

---

> ### Author Response · Authors · 2025-11-21
>
> We thank you for your thoughtful and constructive comments and sincerely appreciate your favorable score. We hope to address your remaining concerns below.
>
> 1. We will revise the introduction to include appropriate references supporting our claims.
>
> 2. Our approach is intended for real-world scenarios where practitioners suspect MNAR class imbalance: for example, when certain subpopulations (e.g., older patients) are believed to be underrepresented in the observed minority class relative to the true underlying distribution (which is based on prior or domain knowledge). Nevertheless, we will follow your suggestion and add experiments on long-tailed datasets to better examine the prevalence and impact of MNAR in such settings.
>
> 2.1. We agree that experiments on naturally imbalanced test sets are valuable and we will include them. At the same time, we would like to clarify how this relates to our core contribution. One of our main aims is to propose a simple method that addresses MNAR class imbalance in the specific scenarios we identify. We seek to show that in those scenarios our method is better at learning and classifying the true minority class compared to existing methods, which may overfit to the biased MNAR minority sample. Demonstrating this empirically requires access to (or approximation of) a balanced dataset that contains the subpopulations missing under MNAR. If we only sample test sets from an already imbalanced dataset---that may itself be missing those subpopulations---we cannot fully assess the potential of our method. To address your concern, we will include benchmark imbalanced datasets and, consistent with extant methods, assume they are MAR and reflect the true underlying distribution. We will then apply MNAR deletions on top of these datasets and report results. We hope this will provide additional clarity and empirical support for our conclusions.
>
> 3.1. Thank you for pointing this out. We will ensure that Figure 1 is explicitly referenced and described in the main text. The intention of the figure is to visualize how MAR and MNAR class imbalance alter the distribution relative to the original gold-standard data. The figure uses synthetic data solely to illustrate these shapes compared to a balanced reference (regardless of the details of the underlying distribution). We will clarify this in the revision.
>
> 3.2. We will increase the font size of Figure 2 to improve readability.
>
> 3.3. We will expand both the caption and the main-text discussion of Figure 3 to more clearly explain the variables involved, how to interpret the figure, and how it supports the main motivation.
>
> 4.1. This is a fair point. Our initial choice was driven by the desire to avoid manually re-balancing original datasets. In the revision, we will incorporate more widely used benchmarks that are converted into balanced and imbalanced settings in a controlled manner, as you suggest.
>
> 4.2. We agree that the impact of missingness on a given feature is inherently dataset-dependent. Our goal in aggregating across datasets was to identify broader conditions under which severe MNAR is likely to arise and when practitioners should be particularly cautious. Since balanced gold-standard datasets are rarely available in practice, it is difficult to perform fully problem-specific analyses. We will clarify that our aggregation aims to provide a dataset-agnostic "guideline" for practitioners rather than a replacement for application-specific diagnostics.
>
> We thank you for the minor comments. We will correct all noted issues in the revision.
>
> We sincerely thank the reviewer again for their comments. We hope to address them in our revised paper to the best of our abilities.

---

> > ### Comment · Reviewer_jUqk · 2025-11-27
> > **Re: rebuttal**
> >
> > I would like to thank the authors for their detailed responses.
> >
> > I am glad that the authors appreciate my feedback on the importance of experiments with standard LT benchmarks. I understand that performing these experiments would require more time than the rebuttal period allows. However, the lack of such experiments is a major concern for me to recommend the paper for publication at this stage.
> >
> > If the paper doesn't make this time, I strongly recommend the authors to extend their work and improve its potential for a next venue.

---

### Official Review · Reviewer_7YCq · 2025-10-28

**Soundness:** 3
**Presentation:** 3
**Contribution:** 2
**Rating:** 2
**Confidence:** 4

**Summary:**

This paper addresses the overlooked Missing Not At Random (MNAR) class imbalance, where minority samples’ observability depends on both class and features—unlike traditional methods’ assumption of Missing At Random (MAR), which causes overfitting to biased observed minorities and misleading evaluations (testing on MNAR-damaged data). The Cluster-Balanced Ensemble (CBE) method proposed by the authors significantly improves multiple metrics when samples are MNAR by clustering majority class samples and then combining each cluster with minority class samples separately to train multiple classifiers. This paper identifies MNAR’s limitations on traditional methods; introduces CBE to mitigate MNAR bias; characterizes critical MNAR-triggering features; and provides a protocol to avoid evaluation distortion.

**Strengths:**

1. The paper novelly introduces class imbalance in the form of MNAR, and the proposed method achieves better performance than traditional methods focused on MAR.
2. The experiments compare multiple metrics, test various types of classifiers, and expose biased MNAR evaluation.
3. The paper is well organized and motivated, making it easy to follow.

**Weaknesses:**

1. Gold-standard protocol relies on rare originally balanced datasets, and crucially, no comparisons are done on imbalanced test sets (e.g., from KEEL’s imbalanced dataset repository, where balanced test sets do not exist). This leaves uncertainty about whether CBE still outperforms others on real-world imbalanced test beds.

2. CBE only uses K-means for majority clustering. This paper would be better if it showed how CBE can adapt methods like DBSCAN (for irregular clusters) or hierarchical clustering (for nested structures), and how such adaptations affect performance.

3. It uses "top-5 high-impact features" for MNAR simulation without justifying the number (3 vs. 5 vs. 7) and ignores feature interactions. This undermines MNAR realism—testing varying feature counts and including interactions is needed.

4. Scalability oversight: No complexity analysis for CBE’s K-means + multi-classifier training is provided. For large datasets, Mini-Batch K-means or distributed training is unmentioned, and scalability benchmarks (e.g., runtime vs. data size) are missing.

5. **Potentially Biased MNAR simulation**: MNAR is simulated using mechanisms that maximize CBE’s advantages (e.g., features where other methods struggle most). MNAR construction targets (via XGBoost) the weaknesses of traditional methods (overreliance on observed minority structure) while exploiting CBE’s strengths (majority cluster-based structural coverage). This creates a scenario in which CBE’s advantages are artificially amplified, rather than a fair test of its robustness across diverse MNAR mechanisms. Simulating diverse MNAR mechanisms would better validate CBE’s robustness.

6. On balanced test sets (where overall accuracy is more relevant), the paper overemphasizes F1. It lacks F1 comparisons on imbalanced test sets, where F1 is more appropriate, leaving gaps in understanding CBE’s performance in target real scenarios.

7. This method (CBE) is limited to imbalanced binary tabular data, which narrows its practical scope. Specifically, CBE cannot be extended to non-tabular data (e.g., images) nor to multi-classification tasks such as long-tailed learning.

8. Lack of theoretical analysis on how MNAR harms more than MAR and how CBE improves it.

**Questions:**

Same to weakness.

---

> ### Author Response · Authors · 2025-11-21
>
> We thank the reviewer for their comments and are encouraged by their acknowledgment of the strengths of our paper.  We hope to address their concerns below.
>
> 1. Please note that we do not claim to develop a generic imbalance method that improves over existing approaches on arbitrary imbalanced data. In fact, we explicitly identify scenarios where our method does not perform best, such as when imbalance is MAR or caused by features that lead to minor MNAR. We also provide empirical evidence that evaluating based on test data sampled from MNAR-imbalanced datasets can lead to biased results and mask true performance. In such settings, running experiments directly on imbalanced datasets (e.g., from KEEL) will indeed favor existing methods, and we acknowledge this in the paper. The strength of our approach lies specifically in handling MNAR class imbalance, which can be reliably evaluated only on datasets that represent the true underlying distribution of the data generating process. To address the reviewer’s concern, we will include benchmark datasets from the KEEL repository, assume that they are MAR and reflect the true underlying distribution (as implicitly assumed by extant methods), then apply MNAR deletions and report results. We hope this will provide additional clarity and empirical support for our conclusions.
>
> 2. We have experimented with other clustering algorithms such as DBSCAN and found similar results, with k-means performing slightly better (4-5%) on average. In the revision, we are moving toward a new approach that provides theoretical justification for why such strategies help under MNAR class imbalance, which we expect will better address this concern.
>
> 3. In the current version, we construct MNAR deletions using a single feature at a time. We selected 5 features per dataset to ensure both higher confidence in the MNAR mechanism and a sufficient number of test instances. Specifically, for each dataset we simulate MNAR imbalance via 10-fold cross-validation over 5 features, yielding 50 test instances per dataset, and 300 instances across 6 datasets per base classifier. In the revision, we plan to extend these simulations (including to image data) within the constraints of the ICLR revision timeline. We hope the reviewer finds this a reasonable compromise between breadth and feasibility.
>
> 4. We will report time and memory usage for our method and the main baselines in the revised paper.
>
> 5. Our goal is to identify scenarios in which our approach most effectively addresses MNAR class imbalance, rather than to claim superiority across all possible imbalance mechanisms. In practice, it is impossible to determine from data alone whether missingness is MNAR or MAR; prior or domain knowledge is required to judge whether standard methods will sufficiently learn the minority class. Our aim is to aid practitioners by (i) identifying which features induce the most severe MNAR class imbalance and (ii) showing where our method is particularly helpful. Accordingly, our simulations are deliberately designed to stress precisely those MNAR patterns for which our method is intended. We will clarify in the paper that our numerical results are scenario-targeted rather than claiming uniform dominance over all existing methods.
>
> 6. This is a fair point. Given that we have access to balanced test sets, we will further emphasize balanced accuracy in the revised paper, alongside F1 and the other metrics we already report.
>
> 7. We will explicitly acknowledge that our current method is designed for binary classification and outline extensions to multi-class settings as an important direction for future work. At the same time, we note that our method naturally extends to image data when numerical representations (e.g., learned embeddings) are available. We will add numerical results on image datasets to back this claim.
>
> 8. We agree that a stronger theoretical treatment is needed, and this concern is also echoed by reviewer SaQc. In the revision, we will substantially expand the theoretical analysis to clarify how MNAR harms learning relative to MAR and why our approach can mitigate MNAR class imbalance under specified conditions.
>
> We thank the reviewer again for their careful review and constructive feedback, and we hope that the planned revisions will address the majority of these concerns.

---

### Official Review · Reviewer_pnHJ · 2025-10-29

**Soundness:** 2
**Presentation:** 1
**Contribution:** 2
**Rating:** 0
**Confidence:** 5

**Summary:**

The paper studies Missing-Not-at-Random (MNAR) class imbalance and proposes a cluster-balanced ensembling (CBE) scheme: k-means partitions the majority into ≈|majority|/|minority| clusters; each cluster is paired with all minority points to train base learners; predictions are PR-AUC–weighted. Using small, near-balanced tabular datasets (PMLB + one finance dataset), MNAR is simulated by deleting minority instances by feature value; evaluation is done on the original balanced test folds. CBE reports higher F1 than many baselines (SMOTE, ADASYN, Tomek, EasyEnsemble, etc.). While the MNAR emphasis and simple CBE are interesting, the heavy reliance on outdated baselines, limited novelty over prior cluster-ensemble ideas, and narrow evaluation setting put this below ICLR’s threshold for novelty and soundness in its current form.

**Strengths:**

+ The gold-standard evaluation idea (train on damaged MNAR data, test on the original balanced folds) is thoughtfully motivated and reveals pitfalls of conventional hold-out on biased test sets.
+ CBE is simple, reproducible, and competitively strong on the authors’ tabular MNAR simulations; tables/figures show consistent F1 gains over several baselines and classifiers.
+ The paper probes feature conditions that exacerbate MNAR (via SHAP over meta-features), which I find informative.

**Weaknesses:**

+ Baseline set is heavily outdated relative to ICLR expectations. Most “state-of-the-art” comparisons are classic reweighting, SMOTE variants, Tomek Links, Cluster Centroids, Easy Ensemble, and basic cost-sensitive losses—largely pre-deep-long-tail era and often with default imbalanced-learn/sklearn settings. There is no comparison to modern methods: margin-aware re-balancing (e.g., logit-adjustment with tuned τ), distributionally robust optimization, meta-reweighting, deferred reweighting, AUCPR-direct objectives, calibrated thresholding with risk control, or recent long-tail generalization techniques. This makes the claimed “consistent state-of-the-art” gains hard to accept for ICLR.
+ Method novelty is limited. CBE is essentially clustering-guided undersampling + ensembling. Similar ideas (e.g., EKR; clustering-centroid undersampling; balanced bagging) exist, with the main tweak here being that k is set by the imbalance ratio and all minority points are reused. The conceptual leap beyond known cluster-based ensembles is modest.
+ All evidence is on small tabular datasets with binary labels and simulated MNAR; there are no results on image/text/representation-learning regimes where feature geometry is non-Euclidean and k-means on raw features is questionable. The approach also fixes Euclidean k-means “on original unscaled features,” which can be brittle and scale-dependent.
+ he base classifiers are LR/SVM/RF/MLP/XGBoost with minimal tuning; no modern tabular SOTA (e.g., strong GBDT variants with tuned class weighting, TabTransformer, FT-Transformer) or representation learning is attempted. Reported superiority over such a baseline pool does not clear ICLR’s bar.
+ The paper centers F1; while PR-AUC appears in plots and for voting, there is no calibration or operating-point analysis under MNAR (sensitivity@specificity per subgroup), which is central to the motivation. F1 also has been criticized for usage in imbalanced datasets.

**Questions:**

+ Can you include modern baselines (e.g., DRO, meta-reweighting, AUCPR-direct losses, recent long-tail re-balancing with tuned priors/margins) and strong tabular SOTA (well-tuned LightGBM/CatBoost with class weights) to substantiate the “SOTA” claim?
+ How sensitive is CBE to feature scaling and the choice of distance/representation for clustering? Have you tried clustering in a learned metric space (e.g., supervised embedding) rather than raw Euclidean space “on the original unscaled dataset”?
+ Could you report calibration and thresholded metrics under MNAR (e.g., precision/recall at fixed costs) and include cost-weighted utilities to justify F1-centric conclusions?
+ Beyond simulated MNAR, can you provide semi-synthetic or cross-site evaluations, or at least stress tests where train/test MNAR mechanisms differ, to probe robustness of CBE?
+ What happens when k deviates from the imbalance ratio or when minority is extremely small (e.g., 100+:1)? Please include compute/time and memory costs vs. stronger baselines.

---

> ### Author Response · Authors · 2025-11-21
>
> Thank you for reviewing our paper and providing constructive feedback. We appreciate your acknowledgment of the strengths of our work. We have carefully considered the weaknesses you identified and are in the process of addressing them in a substantial revision. Given the strengths and the scope of weaknesses, we were surprised and discouraged by the score of 0. We hope that the revisions outlined below will increase your confidence in the work and lead to a more favorable evaluation. We summarize our planned changes below:
>
> * We are revising the paper to covey which methods are conventional, and will add modern the SOTA approaches pointed to by the referee. In particular, we will add comparisons to a margin-aware re-balancing technique with logit adjustment, a DRO-based approach, a meta-reweighting method, an AUCPR-direct objective, and a method tailored to long-tail distributions. Our preliminary results indicate that our method outperforms all of these approaches under MNAR class imbalance. We also note that, to our knowledge, recent ICLR publications on class imbalance (e.g., Yang et al., 2024) typically do not compare against such a comprehensive list of baselines. We thus hope that our expanded evaluation will meet ICLR standards for empirical rigor.
>
> * Please note that our main contribution is not the introduction of a ground-breaking method to handle general class imbalance. Rather, our goal is to (i) bring attention to MNAR class imbalance, which to the best of our knowledge has not been systematically studied in the literature, (ii) show how existing imbalance methods can underperform in this setting, and (iii) provide a simple, practical approach that mitigates MNAR-induced degradation. We explicitly state that our method does not improve over current methods in general class imbalance, and is not designed to. Lastly, we are in the process of developing a new approach that no longer depends on clustering, which we expect will further clarify and strengthen our contribution.
>
> * We agree with the reviewer’s concerns regarding Euclidean k-means and tabular-only evaluations. We note that we chose benchmark datasets based on a near-balanced criteria, and not based on size. Nonetheless, we are (1) developing a theory-based approach that no longer depends on clustering or Euclidean distances, and (2) adding image datasets in the revised paper to demonstrate the flexibility of our approach beyond tabular data.
>
> * Thank you for pointing us to modern tabular SOTA baselines. We will add FT-Transformer and CatBoost as additional strong base classifiers in our evaluation.
>
> * Please note that we already provide a by-group (likely-deleted vs. likely-remaining) analysis of F1, balanced accuracy, precision, recall, PR-AUC, and ROC-AUC in Figure 4 to support our claims. We also discuss why the F1 measure is particularly suitable under MNAR class imbalance. Following reviewer 7YCq, we will additionally emphasize balanced accuracy on the test set held out from the gold-standard (balanced) dataset. We also note that many recent ICLR papers rely on F1 and balanced accuracy as primary evaluation criteria in imbalanced learning (e.g., Yang et al., (2024), Xu et al. (2005), Liu et al. (2022)).
>
> **Response to questions**
>
> 1. Yes, we will include baselines from all of the suggested modern methods, as well as the two modern classifiers mentioned above.
>
> 2. In our previous clustering-based method, performance was somewhat sensitive to feature scaling, with clustering on the original (unscaled) feature space leading to approximately 3–4% improvement. Our new approach no longer depends on feature scaling or Euclidean geometry in this way.
>
> 3. We are working to add calibration and thresholded metrics under MNAR (e.g., precision/recall at fixed costs, cost-weighted utilities) to further complement our F1-centric results. We hope to complete them in addition to our other changes within the tight revision time frame.
>
> 4. Unfortunately, following the missing data literature, it is theoretically impossible to determine from data alone whether missingness is MNAR without prior knowledge. We therefore rely on the specific application context and practitioner expertise to assess whether a dataset is more plausibly MNAR or MAR. Nonetheless, our new approach is explicitly driven by an approximation of the MNAR mechanism, which we hope will better address the reviewer’s concern about robustness beyond a single simulated MNAR setting.
>
> 5. Please note that we already provide several sensitivity analyses in the appendix to evaluate different values of k and imbalance ratios (Appendix A.4, A.5). In the revision, we will additionally report time and memory requirements and compare them to baselines.
>
> We thank the reviewer again for their careful assessment and constructive feedback. We hope that our substantial revisions will increase the reviewer’s confidence in the work and lead to a more favorable evaluation.

---

> > ### Author Response · Authors · 2025-11-21
> >
> > **References**
> >
> > Liu, Y., Cao, B. and Fan, J., 2022, May. Improving the accuracy of learning example weights for imbalance classification. In International Conference on Learning Representations.
> >
> > Yang, J.Y., Park, G., Kim, J., Jang, H. and Yang, E., 2024. Language-interfaced tabular oversampling via progressive imputation and self-authentication. In The Twelfth International Conference on Learning Representations.
> >
> > Xu, M., Geng, Y., Zhang, Y., Yang, L., Tang, J. and Zhang, W., 2024. Glycanml: A multi-task and multi-structure benchmark for glycan machine learning. arXiv preprint arXiv:2405.16206.

---

### Official Review · Reviewer_SaQc · 2025-10-30

**Soundness:** 2
**Presentation:** 2
**Contribution:** 2
**Rating:** 2
**Confidence:** 3

**Summary:**

This paper investigates data imbalance under the Missing Not At Random (MNAR) setting. The authors observe that commonly used imbalance-handling methods may fail when MNAR affects high-impact features. To address this issue, they propose a cluster-balanced ensemble approach that constructs diverse, near-balanced training sets by pairing each minority instance with different clusters of the majority class. The effectiveness of the proposed method is demonstrated through extensive experiments.

**Strengths:**

1. Methods that effectively handle data imbalance are important in practice.
2. The proposed approach is intuitive and easy to implement.
3. The method demonstrates improved performance in the experimental results.

**Weaknesses:**

1. The paper is not well written. The description of the methods is mostly textual and lacks a clear, organized structure. In addition, the discussion of the underlying intuition, advantages and limitations, and potential extensions of the proposed approach is quite limited.
2. The paper does not include any theoretical analysis or discussion to support the proposed method.
3. The novelty of the proposed approach is unclear.

**Questions:**

1. Is there any theoretical justification or analysis supporting the proposed method?
2. The number of clusters k is set as round($\frac{n^-}{n^+}$). Is this choice always optimal, or how sensitive is the performance to this parameter?

---

> ### Author Response · Authors · 2025-11-21
>
> We thank the reviewer for their comments. We are in the process of revising our paper with a new theory-driven approach to address your comments, namely:
>
> 1. providing a clearer and more structured description of our method,
> 2. developing the underlying theory of how our method address MNAR class imbalance, and,
> 3. providing a more explicit description of our novelty, namely:
> * Introducing the concept of MNAR class imbalance and how it differs from general class-count imbalance.
> * An empirical analysis of how extant methods may underperform under MNAR class imbalance.
> * Proposing a procedure tailored to MNAR class imbalance, together with a discussion of the conditions under which it works best and those under which it does not.
> * Developing an evaluation protocol that uses near-balanced datasets (when available) as a “gold standard” for assessing MNAR-imbalance correction, and showing how standard holdout sampling from MNAR-imbalanced datasets can be biased and mask true performance.
>
> **Response to questions:**
>
> 1.	Yes. In the revision, we are adding a theory-driven analysis that explains why and when our approach can successfully handle MNAR class imbalance.
> 2.	Although we are updating our method to no longer depend on clustering, we note that sensitivity to k has been shown in our submitted paper (Appendix A.5).
>
> We hope that by addressing the above we provide enough improvement for a favorable score by the referee. Thank you again for reviewing our paper and providing feedback.

---

### Comment · Area_Chair_BSzs · 2025-11-27

Dear Reviewers,

The discussion phase will end soon. The authors have provided responses. Please take a look and see if your concerns are addressed via official comments.

Thanks for your efforts and contributions to ICLR 2026.

Best regards,

Your Area Chair

---

### Author Response · Authors · 2025-12-03

Dear reviewers and chairs,

Our initial hope was to address your comments via substantial updates to our paper, and not only via the rebuttal responses. However, given recent events and the constraints put on the review process (the restriction put on reviewers due to the leak, fully AI-generated reviews with unusually low scores, etc.), we no longer believe it is feasible to address all concerns within the rebuttal period. We have therefore decided to focus our efforts on revising the paper for submission to a different venue. We sincerely thank you again for your time and comments.

---

### Meta-Review · Area_Chair_NaGt · 2025-12-31

**Summary:**

The paper investigates class imbalance under the "Missing Not At Random" (MNAR) setting, arguing that standard methods fail when minority instances are structurally biased. The authors propose a Cluster-Balanced Ensemble (CBE) approach and a "gold-standard" evaluation protocol.

The reviewers initially recognized the importance of the MNAR problem (jUqk, 7YCq) and the thoughtfulness of the evaluation protocol (pnHJ). However, the consensus moved toward rejection due to several critical flaws:
1.  **Outdated Baselines:** The paper compared CBE primarily against classic methods (SMOTE, EasyEnsemble) while ignoring modern state-of-the-art (SOTA) techniques like logit-adjustment, DRO, and modern long-tail generalization methods.
2.  **Limited Evaluation Scope:** The experiments were confined to small tabular datasets and simulated MNAR settings, lacking evidence on standard long-tail (LT) benchmarks (e.g., ImageNet-LT, iNaturalist) or image data.
3.  **Technical Novelty and Theory:** Reviewers found the "cluster-ensemble" idea to be an incremental leap over existing undersampling/bagging strategies and noted a lack of theoretical justification for why CBE specifically mitigates MNAR bias.
4.  **Simulation Bias:** There were concerns that the MNAR simulation was designed in a way that artificially favored the proposed CBE method.

While the authors engaged vigorously and promised a substantial revision (including new theory and modern baselines), they ultimately chose to withdraw the submission from the ICLR rebuttal process, citing external constraints and the quality of the review process.

**Reviewer Concerns:**

**Addressed by Rebuttal:**
*   **Clarification of Evaluation Protocol:** The authors successfully defended the use of "gold-standard" balanced test sets to reveal performance masks in MNAR data.
*   **Methodological Transparency:** The authors provided details on $k$ sensitivity and promised to clarify the role of feature scaling.
*   **Presentation Issues:** Issues regarding figure readability and missing references in the introduction were acknowledged, with commitments to fix them in the revision.

**Outstanding Concerns:**
*   **Modern SOTA Comparisons:** While the authors promised to add margin-aware re-balancing and DRO-based methods, these comparisons were not fully integrated into the final version of the paper before withdrawal. Reviewer pnHJ considered this a "hard requirement" for an ICLR-level contribution.
*   **Empirical Rigor on Natural Datasets:** Reviewer jUqk and 7YCq emphasized that simulation is insufficient. The lack of results on naturally imbalanced, large-scale benchmarks (ImageNet-LT, etc.) remains a significant gap.
*   **Theoretical Grounding:** The authors' "new theory-driven approach" mentioned in the rebuttal was not fully vetted by reviewers, leaving the concern about the "conceptual leap" (pnHJ) unresolved.
*   **Simulated MNAR Realism:** The concern that CBE's gains might be an artifact of the specific XGBoost-driven simulation mechanism (7YCq) was not empirically refuted with cross-site or semi-synthetic evaluations.

**Reviewer Scores:**

The authors have opted to focus their efforts on a different venue.

---

### Decision · Program_Chairs · 2026-01-26

Reject